# Frequency-Dependent Streaming Potential in a Porous Transducer-Based Angular Accelerometer

**DOI:** 10.3390/s19081780

**Published:** 2019-04-13

**Authors:** Li Ming, Meiling Wang, Ke Ning

**Affiliations:** School of Automation, Beijing Institute of Technology, Beijing 100081, China; mingllove5242@gmail.com (L.M.); jiangyihen@bit.edu.cn (K.N.)

**Keywords:** bandwidth, liquid circular angular accelerometer, low-frequency gain, porous transducer, streaming potential coupling coefficient, surface conductance, transient model, zeta potential

## Abstract

This paper presents a transient model of streaming potential generated when fluid flows through a porous transducer, which is sintered by glass microspheres and embedded in the circular tube of a liquid circular angular accelerometer (LCAA). The streaming potential coupling coefficient (SPC) is used to characterize this proposed transient model by combining a capillary bundle model of a porous transducer with a modified Packard’s model. The modified Packard’s model is developed with the consideration of surface conductance. The frequency-dependent streaming potential is investigated to analyze the effect of structure parameters of porous media and the properties of the fluid, including particle size distribution, zeta potential, surface conductance, pH, and solution conductivity. The results show that the diameter of microspheres not only affects bandwidth and transient response, but also influences the low-frequency gain. In addition, the properties of the fluid can influence the low-frequency gain. Experiments are actualized to measure the steady-state value of permeability and SPC for seven types of porous transducers. Experimental results possess high consistency, which verify that the proposed model can be utilized to optimize the transient and steady-state performance of the system effectively.

## 1. Introduction

Compared with angular displacement and velocity, angular acceleration manifests more efficient characterization of the high-order properties of complex systems. The angular accelerometer based on direct measurement of angular acceleration is widely used in rotation control, navigation, and vibration detection [1]. Recently, a new liquid circular angular accelerometer (LCAA) [2,3,4,5,6] was developed based on inertial liquid mass. Compared with other types of angular accelerometers, such as a molecular electronic transducer (MET) based on four electrodes [7,8,9,10], MEMS [11], heat transfer [12,13], and electromagnetic [14], LCAA possesses a balanced performance within the frequency range, accuracy, and space consumption.

The structure of LCAA was introduced by Cheng [4]. The porous transducer is a critical component of LCAA, which is sintered by glass microspheres under high temperature, and it is the only primary difference when compared with MET-based on four electrodes [7,8,9,10]. According to the principle of LCAA [2], the system of LCAA can be divided into two subsystems including a fluidic system and a molecular electronic system. Although, plenty of works on the fluidic system have been conducted and different models for fluid systems have been proposed [3,4,5], there are still many problems in establishing a theoretical model of the molecular electronic system, which is based on the electrokinetic effect [15] generated when fluid flows through a porous transducer. Laboratory experiments were designed to measure the steady-state streaming potential coupling coefficient (SPC) [15,16,17,18,19]. The theoretical model of SPC in porous media was studied and concluded as the Helmholtz–Smoluchowski equation (H-S equation) [15], giving a linear relationship between streaming potential and applied pressure difference. In addition, researchers also conducted in-depth research on factors affecting streaming potential in porous media, mainly analyzing solid-liquid materials and the macroscopic and microscopic parameters of porous media [15,16,17,18,19]. Several equivalent models of porous media were developed to analyze the influence of structure parameters on the electrokinetic effect, specifically the capillary bundle model [20,21,22] and the pore network model [23,24,25]. In order to obtain the mathematical model of a molecular electronic system in LCAA, the dynamic characteristics of streaming potential are intrinsic parts of the theoretical analysis.

For better understanding of the electrokinetic effect in various porous media, laboratory experiments have been investigated [26,27,28,29], which were used for qualitative analysis. Without the consideration of surface conductance [16,18], Packard [30] derived an expression of transient SPC in a circular tube by utilizing the Navier–Stokes equation. In order to simplify the calculation of the Bessel function, Reppert [31] rewrote Packard’s model based on the thin electrical double layer (EDL) assumption, which was corrected by Tradif [26] later. Pride [32] obtained an expression of transient SPC in complex porous media by combining the Navier–Stokes equation with the Maxwell equation, and it was modified by Tradif [26]. After analyzing the assumptions and constraints of the four proposed models [33], Packard’s model was finally selected.

In this paper, we present a modified Packard’s model considering surface conductance, calculated by Revil’s model [16]. Combining the capillary bundle model of a porous transducer in LCAA [34], the modified Packard’s model is extended to the capillary bundle, and the dynamic model of streaming potential in porous media is established and employed to analyze the influence of structure parameters such as the particle size distribution (PSD), and solution properties like the zeta potential, surface conductance, pH, and solution conductivity on dynamic performance. In addition, experiments to measure steady-state permeability and SPC were actualized for seven types of porous transducers with different PSD. Compared with the permeability predicted by the Kozeny–Carman model [35,36], the permeability estimated by the capillary bundle model possesses higher accuracy, specifically lower than 15%.

## 2. System Structure and Principle of LCAA

The physical prototype and structure diagram of LCAA are illustrated in Figure 1. The main structure [3] is a circular tube made of glass, and the fluid mass flows in this tube. The porous transducer is a critical component of LCAA, which is sintered by glass microspheres and embedded in a circular tube.

The principle of LCAA is shown in Figure 2. The circular tube together with the transducer move with external angular acceleration input. The pressure difference between the ends of the porous transducer results from the relative motion between fluid and transducer. After that, streaming potential is generated due to the EDL on the interface between the liquid and solid. The structure of EDL is illustrated in Figure 3. According to the principle, LCAA can be divided into two parts, specifically a fluidic system and a molecular electronic system. The theoretical model of fluidic system was proposed by Cheng [3], which was used to analyze the influence factors such as wave speed, the structure parameter of the circular tube, and the permeability of the transducer. In this paper, a transient model of the molecular electronic system is developed, and its influence factors are analyzed.

## 3. Theoretical Analysis of the Transient Model of the Electrokinetic Effect

This section concerns three aspects of the transient model in the molecular electronic system based on the electrokinetic effect. Specifically, we modify a transient model of the circular tube, establish a transient model of the molecular electronic system, and analyze its influence factors.

### 3.1. Modifying the Transient Model in the Circular Tube

The electrokinetic effect in a molecular electronic system can be characterized by the streaming potential coupling coefficient $C_{sp}$, which is the ratio of the streaming potential to pressure difference applied to the flow path. The frequency-dependence of $C_{sp}\left(\omega \right)$ has been studied for capillary tubes [26,30,31] and porous media [26,32].

• Packard’s model:

Packard [30] proposed a transient model of streaming potential $E_{sp}\left(\omega \right)$ for a single circular tube by neglecting surface conductance in EDL and charge distribution in the diffusion layer. Based on the Navier–Stokes equation, $E_{sp}\left(\omega \right)$ is given by:(1)$$E_{sp}\left(\omega \right)=E_{0}\left(\frac{2J_{1}\left(kr_{c}\right)}{kr_{c}J_{0}\left(kr_{c}\right)}\right)\Delta P\left(\omega \right),$$
where $E_{0}=\varepsilon \zeta /\mu \sigma_{0}$ is the steady-state streaming potential [30], ΔP(ω) presents the applied pressure difference, ε and μ is the dielectric constant and the dynamic viscosity of the fluid, respectively, ζ means the zeta potential of the bulk fluid, $\sigma_{0}$ is the fluid conductivity, $r_{c}$ denotes the radius of a circular tube, and *k* is given by:(2)$$k=\sqrt{-i\omega \rho /\mu },$$
where $i^{2}=-1$ and ω and ρ are the angular frequency of the external input and the density of the fluid. $J_{n}$ denotes the $n^{\mathrm{th}}$-order Bessel function. The streaming current is given by:(3)$$I_{sp}\left(\omega \right)=I_{0}\left(\frac{J_{1}\left(kr_{c}\right)}{kr_{c}J_{0}\left(kr_{c}\right)}\right)\Delta P\left(\omega \right),$$
where $I_{0}=-2\pi \varepsilon \zeta r_{c}^{2}/\mu L_{c}$, and $L_{c}$ is the actual length of the capillary tube.

• Modified Reppert model:

Packard’s model was rewritten based on the thin EDL assumption, aiming at simplifying the calculation of the Bessel function [31]. After that, it was corrected by Tardif [26], expressed as:(4)$$E_{sp}\left(\omega \right)=E_{0}{\left(1+{\left(-\frac{2}{r_{c}}\sqrt{\frac{\mu }{\omega \rho }}\left(\frac{1}{\sqrt{2}}-i\frac{1}{\sqrt{2}}\right)\right)}^{-2}\right)}^{-1/2}.$$

In order to study the frequency-dependent streaming potential for the porous transducer, Packard’s model is modified as:(5)$$E_{sp}\left(\omega \right)=\frac{\varepsilon \zeta }{\mu \left(\sigma_{0}+\sigma_{s}\right)}\left(\frac{2J_{1}\left(kr_{c}\right)}{kr_{c}J_{0}\left(kr_{c}\right)}\right)\Delta P\left(\omega \right)$$
where $\sigma_{s}$ is the surface conductivity.

### 3.2. Establishing the Transient Model of the Molecular Electronic System

#### 3.2.1. Capillary Bundle Model of the Porous Transducer

The steady-state model of the molecular electronic system is presented by employing the capillary bundle model [34], in which the porous transducer is equivalent to a bundle of circular capillaries with the same tortuosity $\tau_{c}$, as shown in Figure 4. The capillary radius distribution (CRD) can be calculated from PSD, and both are lognormal distributions, respectively presented by $\ln\left(d_{p}\right)∼N\left(\mu_{d},\sigma_{d}^{2}\right)$ and $\ln\left(r_{c}\right)∼N\left(\mu_{c},\sigma_{c}^{2}\right)$ [34]. The parameters of CRD can be derived from PSD [34], specifically as:(6)$$\mu_{c}=\mu_{d}-\ln\left(2\Theta \right),$$
(7)$$\sigma_{c}=\sigma_{d},$$
where $\Theta =\sqrt{m^{2}F^{2}/3}$, *m* is the cementation index of the porous media, and $F=\phi^{m}$ presents the formation factor. ϕ means the porosity of porous media, calculated by [34]:(8)$$\phi =\frac{\pi \tau_{c}}{A}E\left(r_{c}^{2}\right).$$
The steady-state permeability is derived based on the capillary bundle model [34], presented as,
(9)$$K_{0}=\frac{Q\mu L}{A\Lambda P}=\frac{\phi }{8\tau_{c}^{2}}\frac{E\left(r_{c}^{4}\right)}{E\left(r_{c}^{2}\right)}$$
where the expression of the original moment is:(10)$$E\left(r_{c}^{n}\right)=e^{n\mu_{c}+\frac{n^{2}\sigma_{c}^{2}}{2}},n=1,2\cdots$$

#### 3.2.2. Transient Model of the Electrokinetic Effect for the Capillary Bundle

Based on the capillary bundle model, the transient streaming current of the porous transducer can be expressed by:(11)$$\begin{array}{l} {\tilde{I}}_{sp}\left(\omega \right)=\Delta P\left(\omega \right)\cdot \int_{r_{\min}}^{r_{\max}}I_{0}\left(\frac{J_{1}\left(kr_{c}\right)}{kr_{c}J_{0}\left(kr_{c}\right)}\right)n\left(r_{c}\right)dr_{c}=\Delta P\left(\omega \right)\cdot \int_{r_{\min}}^{r_{\max}}\frac{-2\pi \varepsilon \zeta r_{c}^{2}}{\mu L_{c}}\left(\frac{J_{1}\left(kr_{c}\right)}{kr_{c}J_{0}\left(kr_{c}\right)}\right)n\left(r_{c}\right)dr_{c}\\ =\frac{-2\pi \varepsilon \zeta }{\mu L_{c}}\Delta P\left(\omega \right)\cdot \int_{r_{\min}}^{r_{\max}}r_{c}^{2}\cdot \left(\frac{J_{1}\left(kr_{c}\right)}{kr_{c}J_{0}\left(kr_{c}\right)}\right)n\left(r_{c}\right)dr_{c} \end{array}.$$

Considering the dynamic balance of this transient flow, a conduction current is formed to balance the transient streaming current, given by,
(12)$${\tilde{I}}_{sp}\left(\omega \right)=-{\tilde{I}}_{c}\left(\omega \right)=-\Sigma \cdot E_{sp}\left(\omega \right)=\int_{r_{\min}}^{r_{\max}}\left(\frac{\pi r_{c}^{2}\sigma_{0}}{L_{c}}+\frac{2\pi r_{c}\Sigma_{s}}{L_{c}}\right)n\left(r_{c}\right)dr_{c}\cdot E_{sp}\left(\omega \right)$$
where $\Sigma_{s}$ means surface conductance. Thus, the transient streaming potential is expressed by:(13)$$\begin{array}{l} E_{sp}\left(\omega \right)=\frac{{\tilde{I}}_{sp}\left(\omega \right)}{-\Sigma }=\frac{\frac{-2\pi \varepsilon \zeta }{\mu L_{c}}\Delta P\left(\omega \right)\cdot \int_{r_{\min}}^{r_{\max}}r_{c}^{2}\cdot \left(\frac{J_{1}\left(kr_{c}\right)}{kr_{c}J_{0}\left(kr_{c}\right)}\right)n\left(r_{c}\right)dr_{c}}{-\int_{r_{\min}}^{r_{\max}}\left(\frac{\pi r_{c}^{2}\sigma_{0}}{L_{c}}+\frac{2\pi r_{c}\Sigma_{s}}{L_{c}}\right)n\left(r_{c}\right)dr_{c}}\\ =\frac{2\varepsilon \zeta }{\mu }\cdot \frac{\int_{r_{\min}}^{r_{\max}}r_{c}^{2}\cdot \left(\frac{J_{1}\left(kr_{c}\right)}{kr_{c}J_{0}\left(kr_{c}\right)}\right)n\left(r_{c}\right)dr_{c}}{E\left(r_{c}^{2}\right)\sigma_{0}+2\cdot E\left(r_{c}\right)\Sigma_{s}}\Delta P\left(\omega \right) \end{array}.$$

In order to simplify the integral operation of the Bessel function, a uniform distribution of the capillary bundle with the same porosity is utilized, and the equivalent radius of the capillary is obtained by:(14)$$R_{c}=\frac{\int_{r_{\min}}^{r_{\max}}r_{c}n\left(r_{c}\right)dr_{c}}{\int_{r_{\min}}^{r_{\max}}n\left(r_{c}\right)dr_{c}}=E\left(r_{c}\right).$$
The equivalent number of capillaries is given by:(15)$$N_{c}=\frac{E\left(r_{c}^{2}\right)}{E^{2}\left(r_{c}\right)}.$$
Adopting (14) and (15), (13) is rewritten as:(16)Espω=2εζμ·∫rminrmaxrc2·J1krckrcJ0krcnrcdrcErc2σ0+2·ErcΣsΔPω=2εζμ·Rc2J1kRckRcJ0kRcNcErc2σ0+2·ErcΣsΔPω=2εζμ·J1kRckRcJ0kRcσ0+2Σs·ErcErcErc2Erc2ΔPω.

Thus, SPC is derived as:(17)Cspω=EspωΔPω=2εζμσ0+2Σs·ErcErcErc2Erc2J1kRckRcJ0kRc.

### 3.3. Analyzing the Influence Factors of the Electrokinetic Effect

According to (17), the parameters of PSD and the properties of the solution are both influence factors for the calculation of $C_{sp}$, which are analyzed as follows.

#### 3.3.1. Effect of the Structure Parameters of the Porous Transducer

The main parameter of the porous transducer is the PSD of the microsphere, which is obtained by measurement. The CRD of the capillary bundle is derived by (6) and (7). The equivalent mean radius of the capillaries is expressed by (14). In addition, the permeability of porous media not only affects the fluidic system, but also influences the electrokinetic effect in the molecular electronic system. A transient model of permeability [4] is expressed as,
(18)$$K\left(\omega \right)=\frac{K_{0}}{{\left(1-\frac{2i\tau_{c}^{2}\rho \omega }{\pi \mu \Lambda^{2}\phi^{2}}\right)}^{1/2}-\frac{i\tau_{c}K_{0}\rho \omega }{2\pi \mu \phi }}$$
where Λ is the characteristic length of the porous transducer, which can be calculated by the mean diameter ${\bar{d}}_{p}$ of PSD, specifically as:(19)$$\Lambda =\frac{{\bar{d}}_{p}}{3\left(F-1\right)}.$$

Adopting (18), the transition frequency is obtained by:(20)$$\omega_{c}=\frac{\mu }{\rho FK_{0}}.$$

#### 3.3.2. Effect of the Properties of the Solution

Without considering the effect of temperature, three other properties of the solution mentioned in (17) are considered with dependence on the conductivity of the solution, including permittivity, zeta potential, and surface conductance.

Compared with the correlation fitted by Worthington [37], an empirical correlation [38] is derived and utilized to convert fluid conductivity into electrolyte concentration $C_{f}$. This expression is valid for the solution with $C_{f}\in \left(0.0001,0.1\right)$M.
(21)$$\begin{array}{l} \log C_{f}=-1.03024+1.06627\log\left(\sigma_{0}\right)+2.41239\times 10^{-2}{\left(\log\left(\sigma_{0}\right)\right)}^{2}\\ +3.68102\times 10^{-3}{\left(\log\left(\sigma_{0}\right)\right)}^{3}+1.46369\times 10^{-4}{\left(\log\left(\sigma_{0}\right)\right)}^{4} \end{array}$$

The permittivity of electrolyte solution can be calculated by employing the following equation,
(22)$$\varepsilon =8.85\times 10^{-11}\left(8-13C_{f}+1.065C_{f}^{2}-0.03006C_{f}^{3}\right)$$

In addition, the dynamic viscosity was selected to be a constant.

For a brine solution, the dependence of the concentration for the zeta potential and surface conductance was established by Revil [16]. The zeta potential is modeled by,
(23)$$\zeta =\frac{2k_{b}T}{3e}\ln\left[\frac{\sqrt{8\times 10^{3}\varepsilon k_{b}TN_{A}}}{2e\Gamma_{s}^{0}K_{\left(-\right)}}\left(\frac{C_{a}+C_{f}+10^{-pH}}{\sqrt{I}}\right)\left(10^{-pH}+C_{f}K_{Me}\right)\right]$$
in which $k_{b}$ is the Boltzmann constant as $1.381\times 10^{-23}\;J/K$, *T* denotes the temperature of the solution, the elementary electronic charge *e* is $1.602\times 10^{-19}\;C$, Avogadro’s constant $N_{A}$ equals $6.022\times 10^{23}/\mathrm{mol}$, $\Gamma_{s}^{0}$ means the surface site density chosen as $10\;\mathrm{site}/{\mathrm{nm}}^{2}$, $K_{\left(-\right)}$ means the disassociation constant, and $C_{a}$ is the concentration of acid in solution. $I\approx C_{f}$ is the ionic strength of the bulk solution, and $K_{Me}$ is the binding constant for cation adsorption.

Surface conductance is obtained by,
(24)$$\Sigma_{s}=\Sigma_{s}^{EDL}+\Sigma_{s}^{Stern}+\Sigma_{s}^{Prot}$$
and is composed of three parts due to the ionic conductance of EDL ($\Sigma_{s}^{EDL}$), the Stern layer ($\Sigma_{s}^{Stern}$), and proton transfer ($\Sigma_{s}^{Prot}$). Compared with $\Sigma_{s}^{Stern}$ and $\Sigma_{s}^{Prot}$, $\Sigma_{s}^{EDL}$ is small enough to be negligible; while $\Sigma_{s}^{Prot}$ can be selected as $2.4\times 10^{-9}$ S.
(25)$$\Sigma_{s}^{Stern}=\frac{e\beta_{s}\Gamma_{s}^{0}K_{Me}C_{f}}{10^{-pH}+K_{\left(-\right)}\Omega^{2/3}+C_{f}K_{Me}}$$
with:(26)$$\Omega =\frac{\sqrt{8\times 10^{3}\varepsilon k_{b}TN_{A}}}{2e\Gamma_{s}^{0}K_{\left(-\right)}}\left(\frac{C_{a}+C_{f}+10^{-pH}}{\sqrt{I}}\right)\left(10^{-pH}+C_{f}K_{Me}\right).$$

## 4. Experiments

As shown in Figure 5a, a SurPASS electrokinetic analyzer [4] is employed to investigate the hydrodynamic and electrokinetic characteristics of the porous sample. The measuring unit illustrated in Figure 5b is constructed by:Pressure transducers and electrodes;Porous sample;Three-way valves;Syringes for electrolyte transport;Conductivity probe and pH electrode;Cylindrical measuring slot;Reservoir cup.

Seven types of porous transducers with different PSDs were utilized for the test, where the PSDs were controlled by sieves with different sizes. These transducers were made by pouring an amount of glass microspheres in a cylindrical mold and sintered under high temperature. Transducer size and mass were measured, then washed by pure water, and dried in a microwave to avoid the influence of impurities. Finishing the above-mentioned preparation, the transducer was embedded on a cylindrical measuring slot, and solution flowed through it, which was selected as a 0.0115 mol/L sodium chloride solution (NaCl). The structure parameters including CRDs are listed in Table 1, where $\mu_{d}$ and $\sigma_{d}$ are the parameters of the microspheres. The porosity ϕ was obtained by the weighing method, and θ was given by $\sqrt{m^{2}F^{2}/3}$. $\mu_{c}$ and $\sigma_{c}$ are parameters of CRD, which were calculated by (6) and (7) separately. Meanwhile, the parameters of the NaCl solution are included in Table 2. Steady-state SPC $C_{0}$ was directly measured, which was used to calculate the permeability $K_{0}$ of the transducer [4].

## 5. Results and Discussion

In this section, the figures are used to discuss the effect of the porous transducer and electrolyte solution on the electrokinetic process. Meanwhile, the proposed transient model of the molecular electronic system is verified and employed to design the LCAA. Finally, some strategies are given to optimize the transient response and low-frequency gain.

### 5.1. Variation from Porous Transducer

Based on (6) and (7), the parameters of CRDs for different transducers are shown in Table 1. Specifically, the radius distribution of the capillary bundle is illustrated in Figure 6. The density of each radius is dimensionless, which was divided by the maximum value of the density. Thus, the peak value of each curve equaled one.

In order to verify the capillary bundle model of the porous transducer, the permeability predicted by the capillary bundle model was compared with the values estimated by the Kozeny–Carman model [35,36]; while both predicting model were compared with experimental permeability. These results are illustrated in Figure 7. We can conclude that the capillary bundle model possessed higher accuracy when compared with the Kozeny–Carman model. The relative errors of the capillary bundle model for B1–B4 were specifically 10.83%, 9.70%, 11.29%, and 15.26%.

Packard’s model of the capillary was compared with three other models proposed for capillary or porous media. The parameters used in the modeling of transient $C_{sp}$ are presented in Table 3. The frequency dependence of $C_{sp}$ is illustrated in Figure 8. It can be seen that the magnitude-frequency characteristic of related models was basically the same, but the phase-frequency characteristic diverged. The Pride model and Tardif model possessed a leading phase, while the lag phase was more reasonable in the physical system. In addition, the Reppert model was simplified with the assumption of thin EDL.

Moreover, Packard’s model was utilized to estimate $C_{sp}$ for different capillary radii, which is shown in Figure 9. The related parameters are given in Table 4. As shown in Figure 9, the amplitude of $C_{sp}$ was normalized by $\varepsilon \zeta /\mu \sigma_{0}$. Meanwhile, the transition frequency $\omega_{c}$ reduced from $3.0166\;\times \;10^{11}$ Hz–$3.0166\times 10^{5}$ Hz, as the radius of the capillary increased from 0.07–70 μm; while the effective radius of the capillary in porous transducer varied from 3–15 μm, as concluded from Figure 6. The effect on the amplitude of $C_{sp}$ was not obtained from Figure 9 due to neglecting the surface conductance.

Finally, Packard’s model was selected in this paper and is modified with the consideration of surface conductance in the following.

### 5.2. Variation from the Electrolyte Solution

Surface conductance was concerned to establish the proposed transient model for different porous samples. The amplitude of SPC $C_{0}$ was calculated with surface conductance for different samples [39]. A reduced SPC $C_{0}/K_{0}$ is plotted in Figure 10, where $K_{0}$ is the permeability of the related sample. As shown in Figure 10, the solid line presents the values calculated with surface conductance, while the dotted line was obtained without considering surface conductance. The experimental data for different samples were measured by Boleve [39]. We can conclude that Packard’s model [30] overestimated $C_{0}$ for low salinity, especially for pure solvent. According to Figure 10a, the estimation error of the dotted line was about 100% for the electrolyte solution with a conductivity of 0.001S/mol. As the fluid conductivity becomes more than 0.1S/mol, the error can be neglected, which is consistency with Revil [16]. Comparing the results of samples with different mean particle diameters $d_{0}$, the estimated error decreased with the increase of $d_{0}$. Specifically, the error was about 25% for Sample S3 as fluid conductivity was equal to 0.001S/mol, which is illustrated in Figure 10d.

The surface conductivity of sample $\sigma_{s}$ was given by $\sigma_{s}=6\Sigma_{s}/d_{0}$ [39]. The relationship between $\sigma_{s}$ and $d_{0}$ is presented in Figure 11 for different surface conductances. Laboratory experiments were actualized as $\Sigma_{s}=4\times 10^{-9}$ S by Boleve [39], which are also presented in Figure 11. The results show that the surface conductivity of the sample dominated for samples with a small size of particles. Meanwhile, a positive correlation between surface conductivity and surface conductance was observed.

As observed from Revil’s model of zeta potential (23) and surface conductance (25), pH is also an important property for modeling SPC in a molecular electronic system. In this paper, the effect of pH on steady SPC is analyzed, which is illustrated in Figure 12. All other parameters of the model are included in Table 2, except pH and fluid conductivity. The amplitude of zeta potential ζ decreased with increasing fluid conductivity $\sigma_{0}$ or reducing the pH of the solution, which is shown in Figure 12a. Ignoring the surface conductance $\Sigma_{s}$, the steady SPC $C_{0}$ is plotted in Figure 12b and possessed the same trend as the zeta potential, which was also concluded by Glover [40]. Combining the surface conductance model illustrated in Figure 12c, the steady SPC $C_{sp}$ showed a different dependency on pH shown in Figure 12d. We should ensure that the pH of the electrolyte solution remains stable in physical applications.

The zeta potential also contributed to the transient model of SPC, which was investigated with different values for the same transducer B3. The result is illustrated in Figure 13. It is obvious that improving the absolute value of zeta potential was the most effective way to optimize the amplitude-frequency characteristic of $C_{sp}\left(\omega \right)$. Based on the conductivity dependence of the zeta potential as shown in Figure 12a, we needed to select a pure solvent with a larger zeta potential as the fluid mass of LCAA. In addition, the zeta potential effect on transient performance was not achieved by the proposed transient model (17).

The transient model of SPC for the transducer (17) was employed to investigate the influence of the structure parameters of the transducer and the properties of the solution. Meanwhile, the dynamic model of permeability (18) was also analyzed. As shown in Figure 14 and Figure 15, they were different in conductivity. Figure 14 was obtained with $\sigma_{0}=1\times 10^{-7}$ S (like pure water), while Figure 15 was given by $\sigma_{0}=115$ mS used in the experiment. The results show that the amplitude of $C_{sp}\left(\omega \right)$ increased with increasing equivalent diameter of microspheres in the transducer with the same fluid conductivity, resulting in a reduction of the bandwidth of the molecular electronic system. Besides, the amplitude of K(ω) also increased with the increase of the equivalent diameter of microspheres, while it reduced the transition frequency of the permeability in the fluidic system. For the same porous transducer, the amplitude of $C_{sp}\left(\omega \right)$ showed a negative dependence on the conductivity of the electrolyte solution. Meanwhile, the effect on bandwidth cannot be concluded by (17).

The steady-state SPC $C_{0}$ for four types of transducers was directly measured. The relationship between $C_{0}$ and the equivalent radius of capillary is presented in Figure 16a. The experimental results were consistent with the predicted results shown in Figure 15a. Specifically, the steady-state SPC increased with the decrease of capillary radius with the conductivity as $\sigma_{0}=115$ mS. In addition, the transition frequency $\omega_{c}$ was calculated by (20). The relationship between $\omega_{c}$ and the equivalent radius of capillary is illustrated in Figure 16b. The variation trend of $\omega_{c}$ was the same as shown in Figure 15d, which decreased as the increase of the capillary radius. Hence, there was a “trade-off” between the transition frequency and steady state SPC. We should design an appropriate PSD for the porous transducer to improve the performance of LCAA.

Combining the transient model of SPC (17) in the molecular electronic system and the dynamic model of the fluidic system [3], we can optimize the low-frequency gain, bandwidth, and dynamic performance of LCAA. The following strategies can be employed to improve the performance indexes of LCAA as listed in Table 5.
Since wave speed is the most important parameter in a fluidic system, improving the wave speed can extend the bandwidth and optimize the dynamic response, while the low-frequency gain remains the same. At present, it can be achieved only by reducing the gas percentage in fluid and increasing the thickness of the circular tube wall, which both have high technical difficulty.In engineering, we can change the radius of a circular tube to improve the bandwidth. However, there is a “trade-off” between low-frequency gain and the performance of transient response. We need to select a suitable value according the requirement of the application.Adjusting the PSD of the porous transducer, the transient response of the molecular electronic system and fluidic system both can be optimized, while the low-frequency gain of the molecular electronic system is deduced.Reducing the inner radius of the circular tube can improve transient performance.The zeta potential is the key property that can effectively increase the low-frequency gain for the molecular electronic system, which can be adjusted by changing the types of solvent or the conductivity of the electrolyte solution.

## 6. Conclusions

This paper presents a transient model of the electrokinetic effect generated in a molecular electronic system of LCAA. With the consideration of surface conductance, Packard’s model is modified. Combining with the capillary bundle model of the porous transducer, the transient model of the electrokinetic effect is established for the porous transducer. With the application of this model, the effect of the porous transducer and electrolyte solution on dynamic performance is investigated.

Specifically, the low-frequency gain is improved by increasing the effective radius of the capillary, which is obtained by the PSD of the porous transducer or increasing the zeta potential. As for transient performance, it can be optimized by changing the PSD of the porous transducer. We should notice that there is a trade-off between bandwidth and low-frequency gain when adjusting the PSD of the transducer. Thus, we need to design the parameter of the transducer according to the requirements of the application. The experiments of the steady-state SPC and permeability for seven types of transducers were actualized, which verified the capillary bundle model and the proposed transient model. Furthermore, some data given by Boleve [39] were also adopted to investigate the effect from the properties of the electrolyte solution.

Finally, the strategies for optimizing the performance of LCAA are proposed by combining the transient model of the molecular electronic system and the fluidic system. These strategies can be employed to guide the design of LCAA.

## Figures and Tables

**Figure 1 sensors-19-01780-f001:**
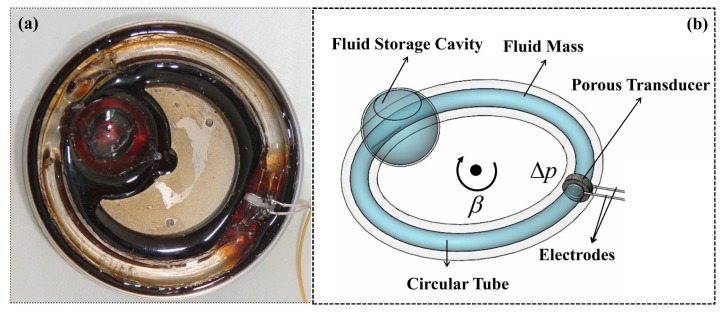
The physical prototype and structure diagram of the liquid circular angular accelerometer (LCAA): (**a**) physical prototype; (**b**) structure diagram.

**Figure 2 sensors-19-01780-f002:**
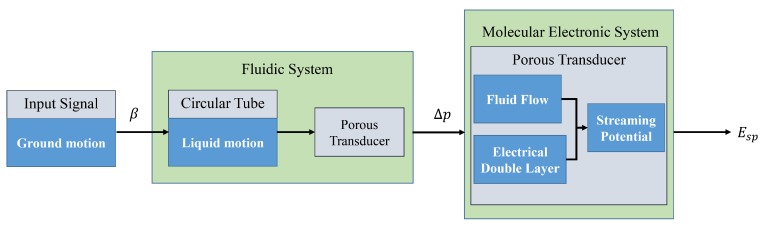
A block diagram of LCAA.

**Figure 3 sensors-19-01780-f003:**
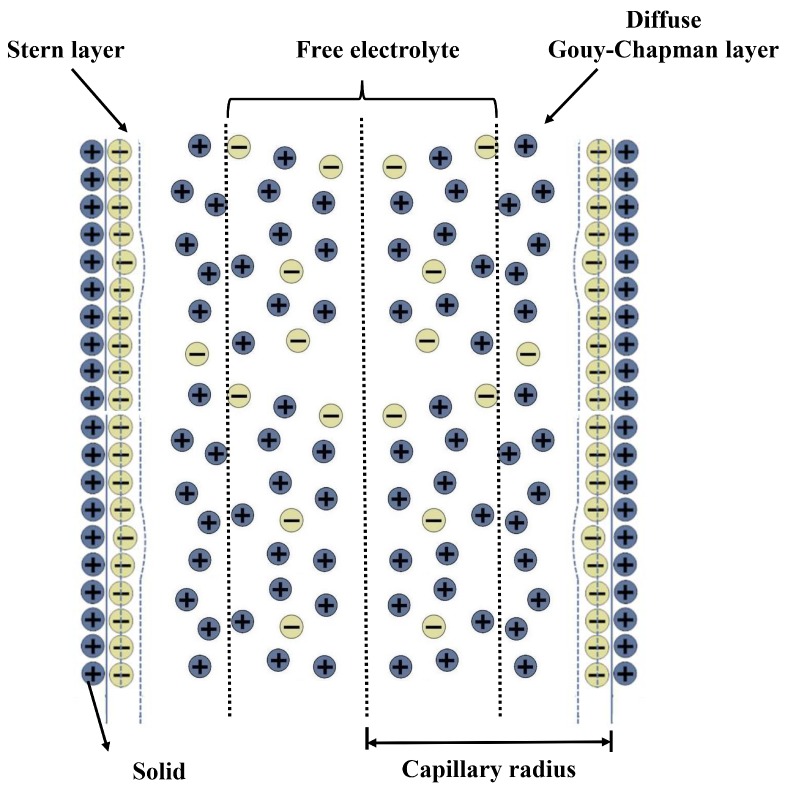
Structure diagram of EDL.

**Figure 4 sensors-19-01780-f004:**
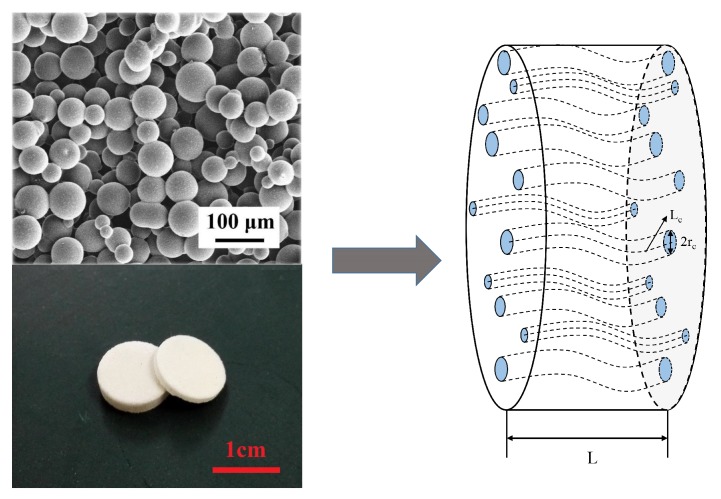
Porous transducers, their microstructure, and the equivalent capillary bundle model.

**Figure 5 sensors-19-01780-f005:**
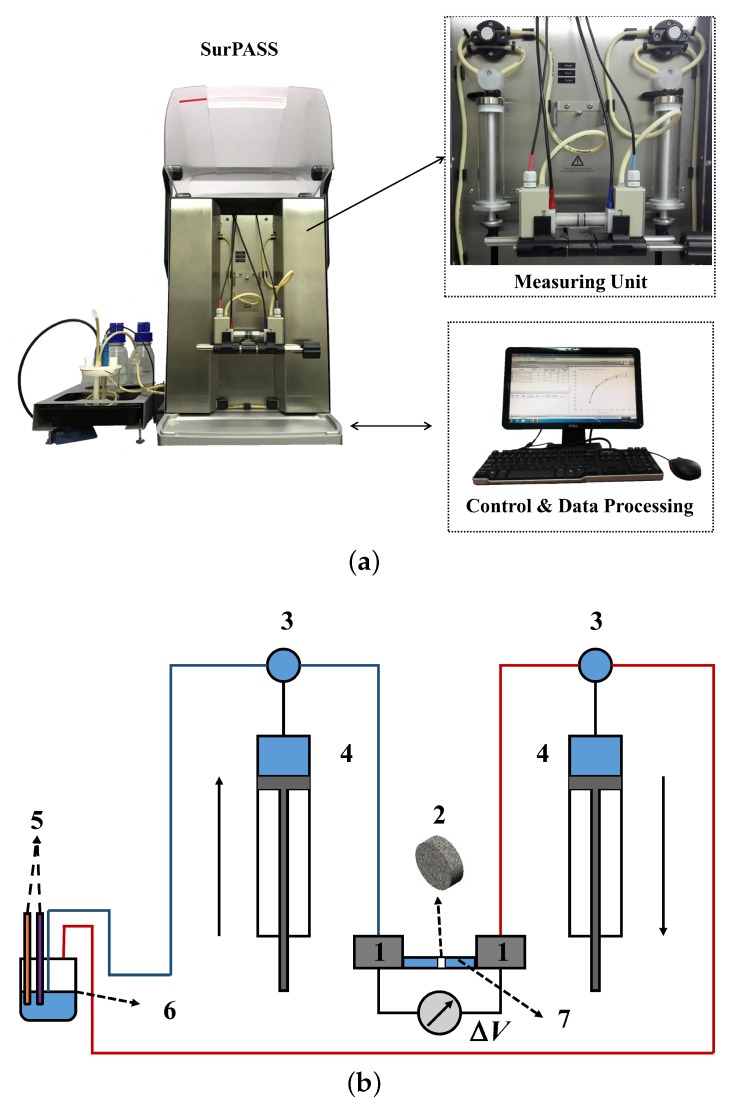
(**a**) Experimental apparatus SurPASS; (**b**) components of the measurement unit in SurPASS.

**Figure 6 sensors-19-01780-f006:**
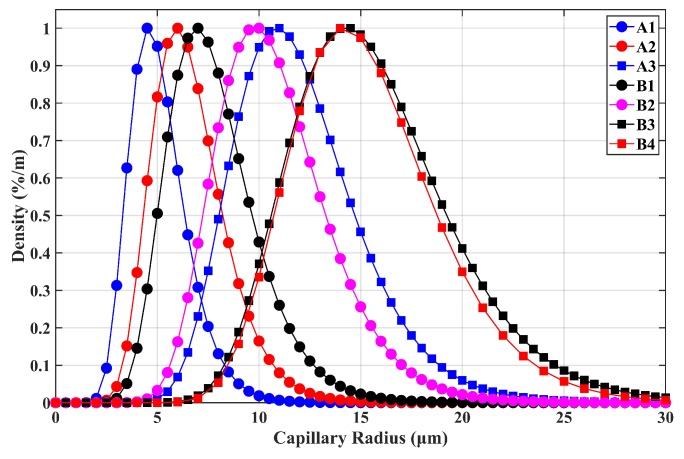
The CRDs of seven types of transducers.

**Figure 7 sensors-19-01780-f007:**
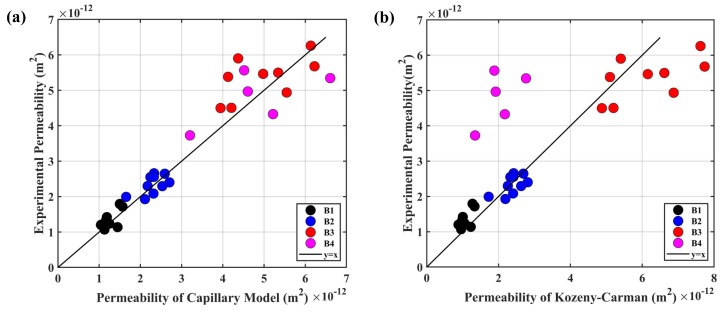
The relationships between the experimental permeability and permeability predicted by different models: (**a**) capillary bundle model; (**b**) Kozeny–Carman.

**Figure 8 sensors-19-01780-f008:**
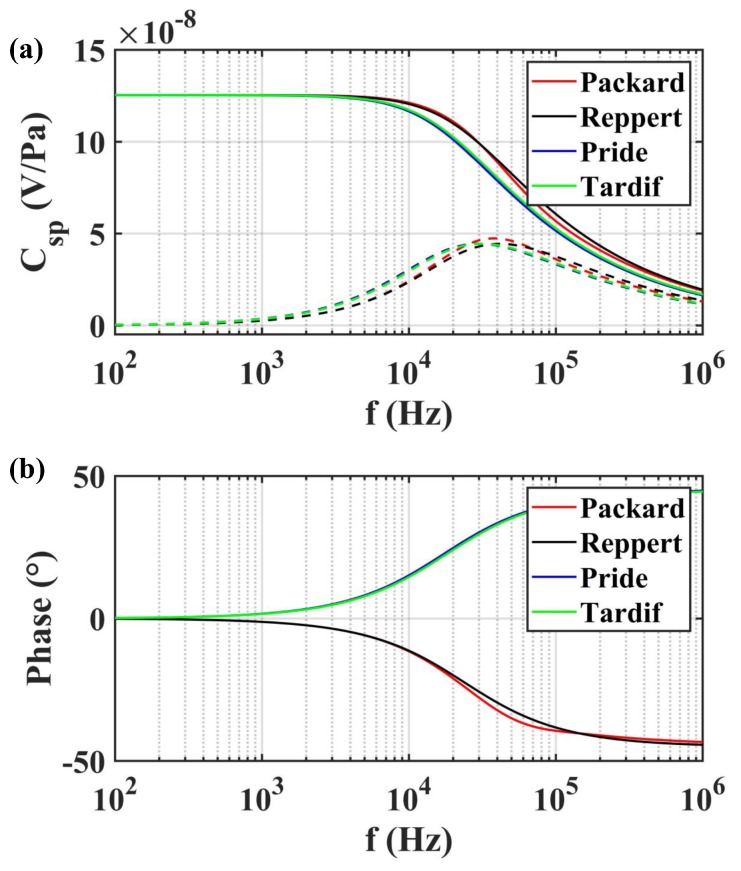
The frequency dependence of streaming potential coupling coefficient (SPC) estimated by different models: (**a**) the amplitude of SPC; (**b**) the phase of SPC.

**Figure 9 sensors-19-01780-f009:**
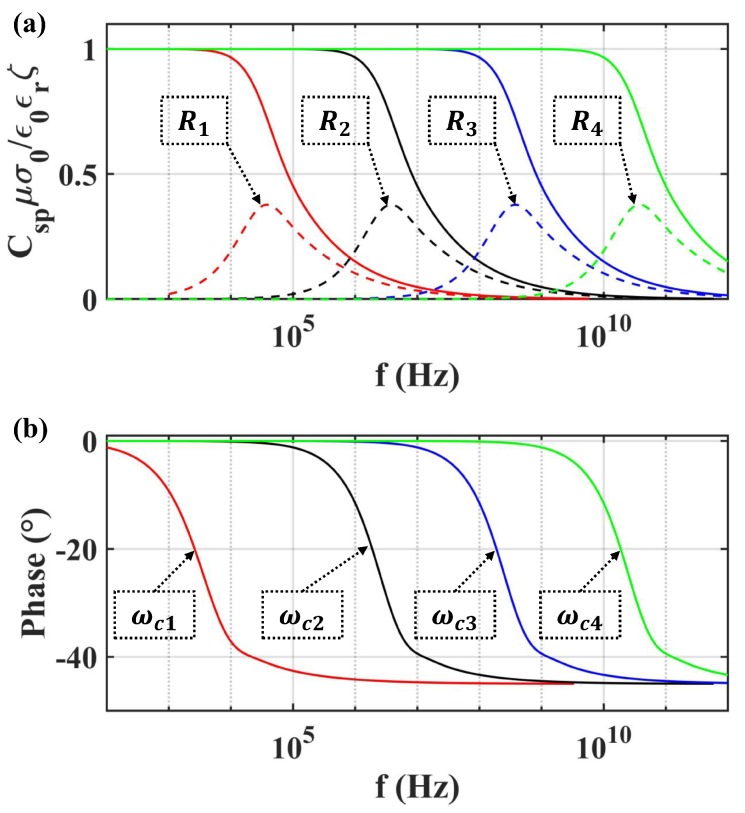
The frequency dependence of the amplitude and phase for SPC with different equivalent radius: (**a**) the amplitude; (**b**) the phase.

**Figure 10 sensors-19-01780-f010:**
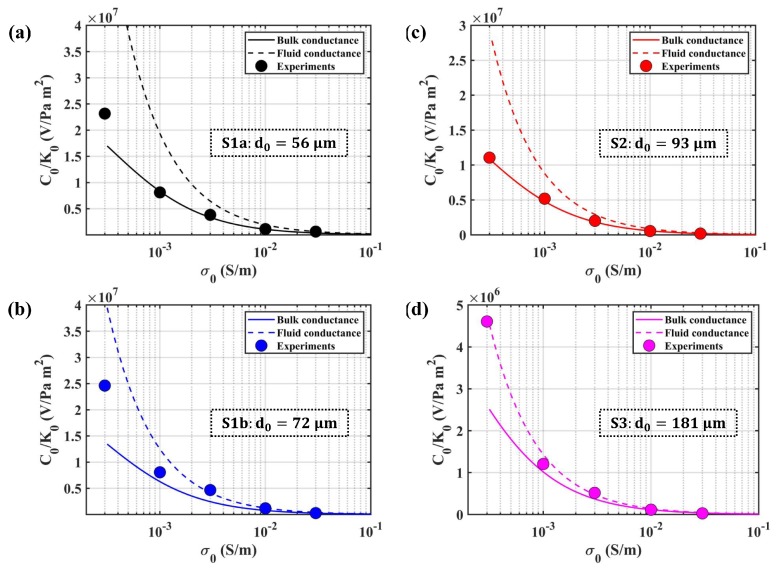
The influence of surface conductance on steady-state SPC. (**a**) reduced steady SPC $C_{0}/K_{0}$ for Sample S1a with the mean diameter as $d_{0}=56$
μm; (**b**) reduced steady SPC $C_{0}/K_{0}$ for sample S1b with the mean diameter as $d_{0}=72$
μm; (**c**) reduced steady SPC $C_{0}/K_{0}$ for sample S2 with the mean diameter as $d_{0}=93$
μm; (**d**) reduced steady SPC $C_{0}/K_{0}$ for Sample S3 with the mean diameter as $d_{0}=181$
μm.

**Figure 11 sensors-19-01780-f011:**
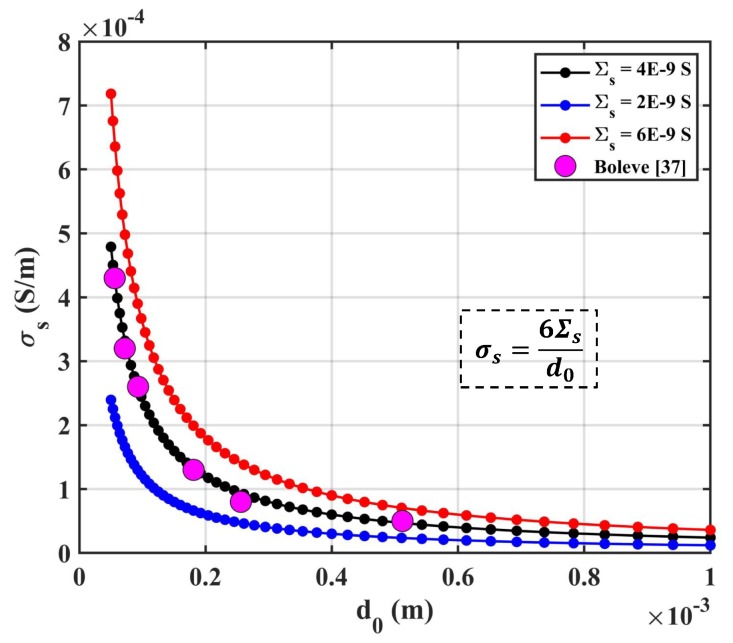
The relationship between $\sigma_{s}$ and $d_{0}$ for different surface conductances: the red line is for $\Sigma_{s}=6\times 10^{-9}$ S; the back line is for $\Sigma_{s}=4\times 10^{-9}$ S; the blue line is for $\Sigma_{s}=2\times 10^{-9}$ S.

**Figure 12 sensors-19-01780-f012:**
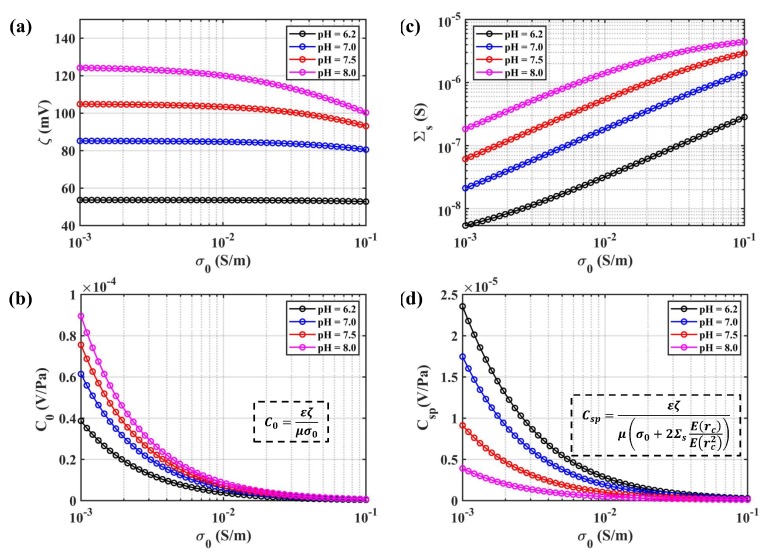
The pH dependence for different properties: (**a**) zeta potential ζ for different pH as a function of fluid conductivity; (**b**) steady SPC without considering surface conductivity $C_{0}$ for different pH as a function of fluid conductivity; (**c**) surface conductance $\Sigma_{s}$ for different pH as a function of fluid conductivity; (**d**) steady SPC with the consideration of surface conductivity $C_{sp}$ for different pH as a function of fluid conductivity.

**Figure 13 sensors-19-01780-f013:**
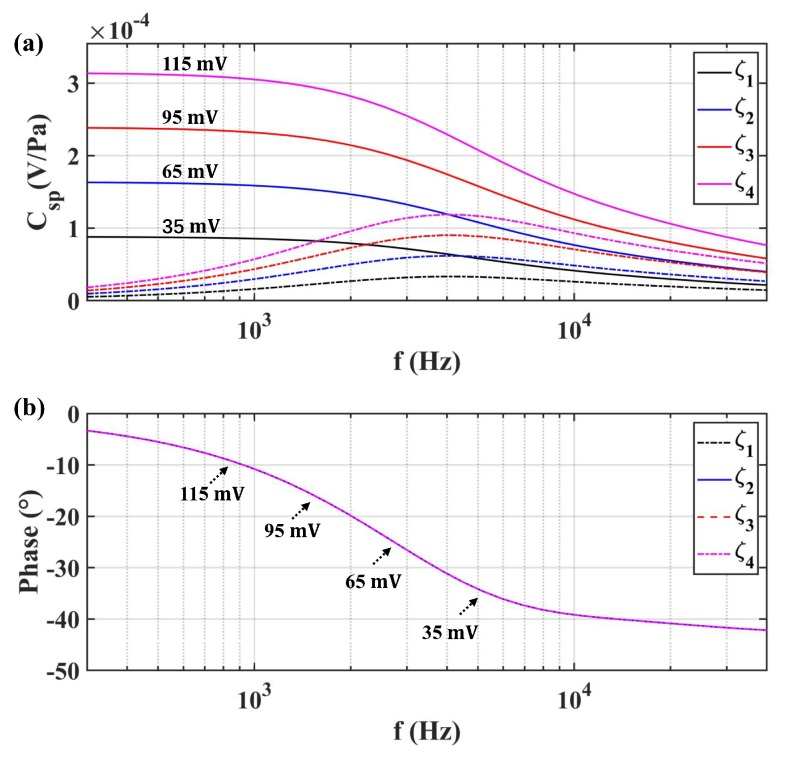
Frequency dependence of $C_{sp}$ with different zeta potentials for B3: (**a**) the amplitude of $C_{sp}$; (**b**) the phase of $C_{sp}$.

**Figure 14 sensors-19-01780-f014:**
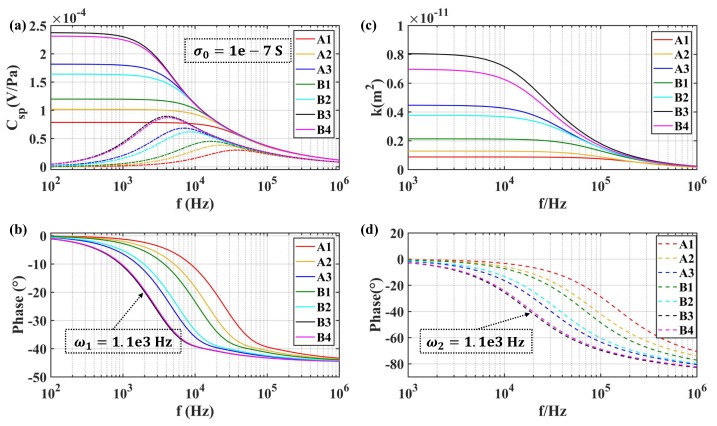
Frequency-dependence of SPC and permeability for different types of transducers at $\sigma_{0}=1\times 10^{-7}$ S: (**a**) the amplitude of $C_{sp}$; (**b**) the phase of $C_{sp}$; (**c**) the amplitude of permeability; (**d**) the phase of permeability.

**Figure 15 sensors-19-01780-f015:**
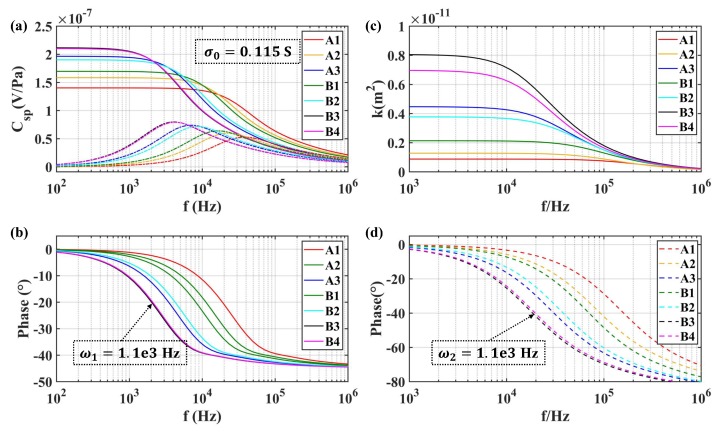
Frequency-dependence of SPC and permeability for different types of transducers at $\sigma_{0}$ = 115 mS: (**a**) the amplitude of $C_{sp}$; (**b**) the phase of $C_{sp}$; (**c**) the amplitude of permeability; (**d**) the phase of permeability.

**Figure 16 sensors-19-01780-f016:**
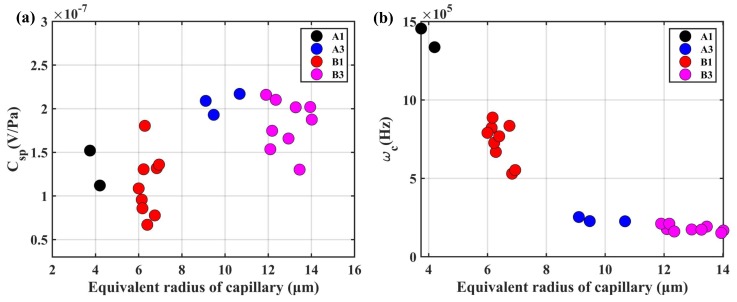
The experimental results for four types of transducers: (**a**) the relationship between the equivalent capillary radius and steady-state SPC; (**b**) the relationship between the equivalent capillary radius and transition frequency.

**Table 1 sensors-19-01780-t001:** Related parameters of porous transducers.

Type	$\mu_{d}$	$\sigma_{d}$	ϕ	θ	$\mu_{c}$	$\sigma_{c}$
A1	−10.521	0.277	0.363	2.728	−12.218	0.277
A2	−10.088	0.273	0.319	3.242	−11.957	0.273
A3	−9.645	0.257	0.358	2.787	−11.363	0.257
B1	−10.156	0.281	0.378	2.587	−11.799	0.281
B2	−9.795	0.257	0.370	2.660	−11.467	0.257
B3	−9.455	0.253	0.380	2.573	−11.093	0.253
B4	−9.376	0.237	0.354	2.823	−11.107	0.237

**Table 2 sensors-19-01780-t002:** Related parameters of the NaCl solution in the experimental test.

Symbol	Value	Symbol	Value
$C_{f}$	0.0115 mol/L	$\Gamma_{s}^{0}$	9.3 $\mathrm{site}/{\mathrm{nm}}^{2}$
$C_{a}$	0	$K\left(-\right)$	$10^{-7.40}$
$\sigma_{0}$	115 mS/m	$K_{Me}$	$10^{-5.50}$
pH	6.2	$\beta_{s}$	$0.4\times 10^{-5}\;m^{2}/\mathrm{sV}$

**Table 3 sensors-19-01780-t003:** The parameters used in modeling $C_{sp}$ with different methods in Figure 8.

Parameter	Value	Parameter	Value
$C_{f}$	0.01 mol/L	*m*	1.5
*T*	300 K	$r_{c}$	$\sqrt{8K_{0}}$μm
ϕ	0.35	${\bar{d}}_{p}$	70 μm
$K_{0}$	$\phi^{3}{\bar{d}}_{p}^{2}/\left(150{\left(1-\phi \right)}^{2}\right)$	ζ	20 mV
$\tau_{c}$	$\phi /\left(1-{\left(1-\phi \right)}^{2/3}\right)$	−−	−−

**Table 4 sensors-19-01780-t004:** The radius of the capillary and related transition frequency used in modeling $C_{sp}$ in Figure 9.

Parameter	1	2	3	4
*R* (μm)	70	7	0.7	0.07
$\omega_{c}$ (Hz)	$3.0166\times 10^{5}$	$3.0166\times 10^{7}$	$3.0166\times 10^{9}$	$3.0166\times 10^{11}$

**Table 5 sensors-19-01780-t005:** Performance indexes of the liquid circular angular accelerometer (LCAA) [3].

Index	Value	Unit
Bandwidth	0.5∼120	Hz
Measurement Range	−25,000∼+25,000	${}^{{}^{\circ}}/s^{2}$
Scale Factor	0.5	$\mathrm{mV}\;s^{2}{/}^{{}^{\circ}}$
Power Supply	±15	V
External Size	Φ75 × 41	mm
Temperature Range	−40∼+60	°C

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
