# Peer review of "Frequency-Dependent Streaming Potential in a Porous Transducer-Based Angular Accelerometer"

_sensors, 2019, doi:10.3390/s19081780_

Round 1

Reviewer 1 Report

Good article, I recommend to accept it.

I would like to invite the authors to clarify two small questions.

1. In their previous article (DOI: 10.1109/JSEN.2016.2628039), the authors predicted and experimentally measured a peak in the transfer function of the porous transducer at the high frequency. How was this peak corrected and a smooth frequency response was obtained?

2. In their previous article (DOI 10.1016/j.sna.2017.02.014), the authors have established dependencies for the permeability of a porous transducer. Why they decided to use the equivalent capillary bundle model? How do these two models correlate with each other?

Author Response

Response to Reviewer 1 Comments

Dear Reviewer

We appreciate your careful reading and instructive comments. Your comments really help us improve the quality of our manuscript. All of them were taken into consideration and we modified them as possible as we can. As for some Comments that we really cannot improve for the non-academic reason, we clarify the reasons elaborately. We hope the current manuscript could be satisfactory. The corresponding modifications and responses to your comments are listed as follows. Besides, the language has been refined in this revision and all modifications have been marked in red font in the manuscript.

Comment 1: In their previous article (DOI: 10.1109/JSEN.2016.2628039), the authors predicted and experimentally measured a peak in the transfer function of the porous transducer at the high frequency. How was this peak corrected and a smooth frequency response was obtained?

Response 1: As shown in Figure 2, the liquid circular angular accelerometer (LCAA) includes fluidic system and molecular electronic system. The previous article (DOI: 10.1109/JSEN.2016.2628039) proposed a dynamic model for the fluidic system. Meanwhile, the influences of several structure parameters, such as the radius of circular tube, not including the structure parameters of porous transducer, were analyzed. The mentioned peak of frequency response is related to wave speed of fluid and the radius of circular tube. However, this manuscript is aimed at developing a transient model for molecular electronic system, which is based on electrokinetic effect. This manuscript investigates the influences of structure parameters of porous transducer and properties of fluid on dynamic streaming potential coupling coefficient. Hence, they are different in the principle. The smooth frequency response is not obtained by correcting the mentioned peak.

Comment 2: In their previous article (DOI 10.1016/j.sna.2017.02.014), the authors have established dependencies for the permeability of a porous transducerWhy they decided to use the equivalent capillary bundle model? How do these two models correlate with each other?

Response 2: The previous article (DOI 10.1016/j.sna.2017.02.014) established a steady-state permeability model of the mentioned porous transducer. The proposed model was employed to investigate the influences of structure parameters of porous transducer. The permeability dependency of steady-state streaming potential coupling coefficient was also analyzed. The mentioned porous transducer was sintered by glass microspheres resulting in complex flowing paths, which was difficult to obtain the accurate distribution of fluid velocity and electrical charge. Thus, the capillary bundle model was developed to be an equivalent model of the porous transducer. The capillary radius distribution was calculated from the particle size distribution of the porous transducer. The capillary bundle model was also employed to estimate the steady-state permeability and streaming potential coupling coefficient. As mentioned in the previous article (DOI: 10.1016/j.ijheatmasstransfer.2017.10.128), plenty of experiments were actualized to validate the capillary bundle model. In this manuscript, the capillary bundle model is employed to establish the transient model of molecular electronic system.

Reviewer 2 Report

In this paper, the authors present a modified Packard’s model with considering surface conductance, calculated by Revil’s model. Combining capillary bundle model of porous transducer in LCAA, the modified Packard’s model is extended to capillary bundle, and dynamic model of streaming potential in porous media is established and employed to analyze the influence of structure parameters and solution properties on dynamic performance. Experimental results show a high consistency with predicted values derived from the model.  

This manuscript is well written and prepared, and the work is good. I think it can be accepted in present form. 

Thanks.

Author Response

Response to Reviewer 2 Comments

Dear Reviewer

We appreciate your careful reading and the recognition of our work. All modifications have been marked in red font in the manuscript. We hope the current manuscript could be satisfactory.

Reviewer 3 Report

Line 18 to 20: Not clear, incorrect grammar, please rephrase. 

Line 23: It should be MEMS
Line 23: Delete “and so on”

Line 29: “work” cannot be investigated, it can be “conducted”

Line 40: refrain from using “and so on”

Line 58: This is a conclusion statement, and not an introduction. Also, please quantify the consistency.

Figure 1: rephrase the caption

The manuscript requires to undergo extensive English editing. 

Line 61: the tube is flowing through the fluid mass?

Section 3.1: Surface conductance is still neglected here.

Sections 3.2 and 3.3: The objective behind presenting the already proven equations is unclear.

Table 1: what are these presented parameters. How they were controlled?

Lines 139 to 144: The reason(s) behind this behavior is unclear.

How the model and Section 4 results compare? The conclusion is unclear.

The novelty of the work is unclear.

What are the conclusions from Figs 13, 14 and 15? Please explain.

Author Response

Response to Reviewer 3 Comments

Dear Reviewer

We appreciate your careful reading and instructive comments. Your comments really help us improve the quality of our manuscript. All of them were taken into consideration and we modified them as possible as we can. We thank for your suggestions about the language and we have corrected all of the language errors that you mentioned. As for some Comments that we really cannot improve for the objective reason, we clarify the reasons carefully. We list the corresponding modifications and responses to your comments in the following. All modifications have been marked in red font in the manuscript. We hope the current manuscript could be satisfactory.

Comment 1: Line 18 to 20: Not clear, incorrect grammar, please rephrase.

Response 1: Thanks for this comment. The vocabulary and expression in the part were not good enough in previous version. We refined the language of this part and we hope that it is of better quality now.

Comment 2: Line 23: It should be MEMS.

Response 2: Thanks for this comment. It’s revised as “MEMS”.

Comment 3: Line 23: Delete “and so on”.

Response 3: Thanks for this comment. The words “and so on” are deleted.

Comment 4: Line 29: “work” cannot be investigated, it can be “conducted”.

Response 4: Thanks for your suggestion and we used an inappropriate word here. The “investigated” is replaced by “conducted”.

Comment 5: Line 40: refrain from using “and so on”.

Response 5: Thanks for this comment. It’s deleted.

Comment 6: Line 58: This is a conclusion statement, and not an introduction. Also, please quantify the consistency.

Response 6: Thanks for this comment. We used an inappropriate expression and did not present a quantified conclusion in the previous version. Now, we modified it and also added one figure as shown in Figure 7 to verify the capillary bundle model. We hope the current expression could be satisfactory.

Comment 7: Figure 1: rephrase the caption. The manuscript requires to undergo extensive English editing. 

Response 7: Thanks for this comment. We have rephrased the caption of Figure 1. For the language expression of this manuscript, we have tried our best to refine it and we hope that would be better now.

Comment 8: Line 61: the tube is flowing through the fluid mass?

Response 8: Thanks for this comment and we used the wrong expression. And this sentence is rephrased to be “The main structure [3] is a circular tube made of glass, and the fluid mass flows in this tube.” We hope it be better now.

Comment 9: Section 3.1: Surface conductance is still neglected here.

Response 9: Thanks for your comment. I presented the modified Packard’s model with the consideration of surface conductance given by equation 5, which is employed to establish the transient model of streaming potential for the porous transducer. The effect of surface conductance was analyzed in Section 5. The results were included in Figure 10.

Comment 10: Sections 3.2 and 3.3: The objective behind presenting the already proven equations is unclear.

Response 10: Thanks for this comment. We simplified the already proven equations. These equations were employed in Section 4 and Section 5 to calculate the related parameters. In the analysis of the results, these equations were used multiple times. Therefore, in order to enhance the readability of this manuscript, we still kept these simplified formulas and omitted their specific derivation process. We hope it would be appropriate.

Comment 11: Table 1: what are these presented parameters. How they were controlled?

Response 11: Thanks for this comment. We added the explain of these presented parameters in Section 4, which were calculated by equations in Section 3. Since these porous transducers were made by pouring amount of glass microspheres in cylindrical mold and sintered under high temperature, we can control these parameters by employing sieves with different sizes to adjust the PSD of glass microspheres.

Comment 12: Lines 139 to 144: The reason(s) behind this behavior is unclear.

Response 12:  Thanks for this comment. In this section, we add one figure to verify the capillary bundle model. The result shows that the permeability predicted by capillary bundle model possesses higher accuracy when compared with Kozeny-Carman model. Both predicting model are compared with experiments for different types of transducers. For the original Figure 7, we have changed it to be Figure 16 in this manuscript to explain the experimental results. Specifically, it is shown in Lines 215 to 223.

Comment 13: How the model and Section 4 results compare? The conclusion is unclear. The novelty of the work is unclear.

Response 13:  Thanks for this comment. We compared four different transient model in Section 5 as shown in Figure 8. We analyzed the results and explained the reasons of choosing Packard’s model. For details, please see Lines 155 to 160. After that we adapted the Packard’s model to study the effect of radius of single capillary, which shown in Figure 9. Moreover, the effect of surface conductance was discussed in Lines 171 to 183 and illustrated in Figure 10. For the influence of pH and zeta potential, the results were separately presented in Figure 11 and Figure 12. The analysis of these results were included in Lines 189 to 203. The proposed transient model was utilized to study the influences of transducer structure parameters and solution properties. The conclusion was included in Lines 204 to 214. In order to verify this proposed model, the experiment was conducted and the results were illustrated in Figure 16. As shown in Figure 16, the results were consistent with the predicted results concluded in Figure 15. The liquid circular angular accelerometer (LCAA) is a new sensor based on directly measurement of angular acceleration. We firstly proposed the transient model of electrokinetic process for LCAA. We also conducted plenty of experimental measurements and simulations to analyze the influences of porous transducer and solution. Based on this presented model, we proposed some technical strategies to improve the performance of LCAA, specifically included in Lines 228 to 241. We hope that the innovations and works of this article could be accepted and recognized.  

Comment 14: What are the conclusions from Figs 13, 14 and 15? Please explain.

Response 14: Thanks for this comment and we did not arrange the position of text description and corresponding figures well, resulting in poor readability. To improve the readability of this manuscript, we adjusted the order of figures and their conclusions to make sure that the conclusions followed the figures. Specifically, the conclusion of Figure 13 is located in Lines 198 to 203. The analysis of Figure 12 and 13 is included in Lines 204 to 214. We hope these modifications would be better.

Round 2

Reviewer 3 Report

Editing the English language is suggested.